# Evidence for dynamic kagome ice

E. Lhotel[1], S. Petit[2], M. Ciomaga Hatnean [3], J. Ollivier[4], H. Mutka[4], E. Ressouche[5]
M.R. Lees [3] & G. Balakrishnan [3]

The search for two-dimensional quantum spin liquids, exotic magnetic states remaining disordered down to zero temperature, has been a great challenge in frustrated magnetism over the last few decades. Recently, evidence for fractionalized excitations, called spinons, emerging from these states has been observed in kagome and triangular antiferromagnets. In contrast, quantum ferromagnetic spin liquids in two dimensions, namely quantum kagome ices, have been less investigated, yet their classical counterparts exhibit amazing properties, magnetic monopole crystals as well as magnetic fragmentation. Here, we show that applying a magnetic field to the pyrochlore oxide $Nd_2Zr_2O_7$, which has been shown to develop three-dimensional quantum magnetic fragmentation in zero field, results in a dimensional reduction, creating a dynamic kagome ice state: the spin excitation spectrum determined by neutron scattering encompasses a flat mode with a six arm shape akin to the kagome ice structure factor, from which dispersive branches emerge.

[1] Institut Néel CNRS, Université Grenoble Alpes, 38042 Grenoble, France. [2] Laboratoire Léon Brillouin, CEA CNRS Université Paris Saclay, CE-Saclay, 91191 Gif-sur-Yvette, France. [3] Department of Physics, University of Warwick, Coventry CV4 7AL, UK. [4] Institut Laue Langevin, F-38042 Grenoble, France. [5] INAC, CEA, Université Grenoble Alpes, CEA Grenoble, F-38054 Grenoble, France. Correspondence and requests for materials should be addressed to E.L. (email: elsa.lhotel@neel.cnrs.fr) or to S.P. (email: sylvain.petit@cea.fr)

The two-dimensional kagome and three-dimensional pyrochlore structures are low connectivity lattices based on corner sharing triangles or tetrahedra, respectively. They form a rich playground to study unconventional magnetic states, such as spin liquids, induced by geometrical frustration. At first glance, they bear no relation to each other, especially when considering their dimensionality. Nevertheless, along [111] (and symmetry-related directions), the pyrochlore lattice can be viewed as a stacking of triangular and kagome layers, as illustrated in Fig. 1a. As a result, if one is able to decouple these layers, two-dimensional physics characteristic of the kagome lattice can develop on the pyrochlore lattice.

This is of specific interest in the context of spin-ice, an original state of matter made of an assembly of Ising spins aligned along the local $\langle 111 \rangle$ directions (which join at the center of the tetrahedra) and coupled by a ferromagnetic interaction[1]. Spin-ice is a degenerate state where the spins locally obey, on each tetrahedron, the 2 in $-2$ out ice-rule, i.e., two spins point in and two spins point out of each tetrahedron. Two-dimensional kagome physics is observed in spin-ice when a magnetic field is applied along the [111] direction: the spins in the triangular planes having their easy axis parallel to the field, they are easily polarized, and thus decouple from the kagome layers. Provided the field is not too strong, a degeneracy persists within the kagome planes, characterized by the kagome ice-rule[2], i.e., 2 spins point into each triangle, and 1 out, or vice versa, as shown in Fig. 1b. This corresponds to the two-dimensional kagome ice state, extensively studied in artificial lattices[3,4] and recently realized in a bulk material[5]. In this state, the algebraic correlations within the tetrahedra characteristic of the spin ice state[6,7] become two-dimensional within the kagome planes[8,9]. They give rise in both cases to a diffuse neutron scattering signal exhibiting pinch points, but with different patterns[10,11]. This field-induced 3D–2D reduction also manifests itself as a magnetization plateau at 2/3 of the saturation magnetization[12].

The way this classical picture is affected by quantum effects, and especially the conditions under which a quantum kagome ice state could be stabilized from a quantum spin ice state has aroused great interest[13–15]. For instance, this issue has been tackled in quantum spin-ice candidates like $Tb_2Ti_2O_7$[16–18] and the Pr pyrochlores[19,20], through the search for magnetization plateau. In this work, we address the kagome ice physics in a

different way by focusing on the effect of a [111] field on a dynamic spin ice state. Our starting point is the peculiar behavior of the $Nd_2Zr_2O_7$ quantum pyrochlore magnet, where classical spin ice physics is considerably modified by the existence of transverse terms in the Hamiltonian: in zero field, the ground state exhibits an all in–all out magnetic structure, while spin ice signatures are transferred in the excitation spectrum, taking the form of a flat spin ice mode[21]. Applying a [111] magnetic field, we show that the flat mode persists and that a dimensional reduction occurs in the excitation spectrum: above about 0.25 T, a flat kagome ice like mode forms in the excitation spectrum, featuring a dynamic kagome ice state. Mean-field calculations using the XYZ Hamiltonian[22] adapted for the $Nd^{3+}$ ion allow us to refine a set of exchange parameters. Some discrepancies between our observations and the calculations, however, point to the existence of more complex processes.

## Results

**Field-induced magnetic structures.** The [111] field-induced phase diagram in $Nd_2Zr_2O_7$ has been probed by magnetization measurements[23,24]. A small anomaly attributed to a magnetic transition is observed at $\mu_0 H_c \approx 0.08$ T, with a hysteretic behavior, on top of a smooth evolution which is not expected for conventional Ising spins. Bragg peak intensities measured by neutron diffraction also show a hysteretic behavior and a discontinuity at $H_c$, confirming that the cusp in the derivative of the magnetization $dM/dH$ corresponds to a change in the magnetic structure (see Fig. 2a, b). The value and orientation of the magnetic moments obtained from the magnetic structure refinements (see Supplementary Note 1) are shown in Fig. 2c, d. Over the whole field range, the spins lying in the kagome planes adopt a 3 in $-3$ out configuration. In a large enough magnetic field, typically 1 T, both types of spins, i.e., the apex and the kagome spins, are saturated, forming the expected ordered classical structure 3 in $-1$ out/1 in $-3$ out. Starting from $-1$ T and sweeping the field up, the ordered components progressively decrease. The apical spin totally loses its ordered moment at about $-0.1$ T, before flipping to the zero field configuration, an all in–all out structure with a partially ordered moment of 0.8 $\mu_B$ (to be compared to 2.3 $\mu_B$, the magnetic moment of the ground doublet)[23]. When further increasing the field, the kagome spins flip at $H_c$ to accommodate the field, and the system returns to a 3 in $-1$ out/1 in $-3$ out structure. Finally, at larger fields, the ordered magnetic moments continue increasing towards the saturated value.

These field-induced magnetic structures qualitatively agree with the conventional behavior of an all in–all out system in a [111] magnetic field. Nevertheless, only a fraction of the expected Nd moment is involved in the ordered magnetic moment in the low-field region, and the magnetization increases smoothly. This is due to the peculiar dipolar octupolar nature of the ground-state Nd doublet[22], which makes the magnetic moment different from a classical Ising spin and allows for non-magnetic transverse components. This results in exotic dynamics, that we have probed by inelastic neutron scattering measurements.

**Evidence for a dynamic kagome ice mode.** In zero field, as previously mentioned, a dynamic spin ice mode is observed[21] at an energy of about 70 $\mu$eV. In the scattering plane perpendicular to [111], this mode is characterized by the star-like pattern shown in Fig. 3a, with a strong intensity around $\mathbf{q} = \mathbf{0}$. On increasing the field, one could expect this feature to disappear at $H_c$. However, the star-like pattern persists up to 0.25 T, where it changes into a pattern with a new structure, at about the same energy, as shown in Fig. 3b: six arms appear, while the scattering intensity decreases around $\mathbf{q} = \mathbf{0}$ and the $(2, -2, 0)$ $\mathbf{q}$-vectors. Upon increasing the

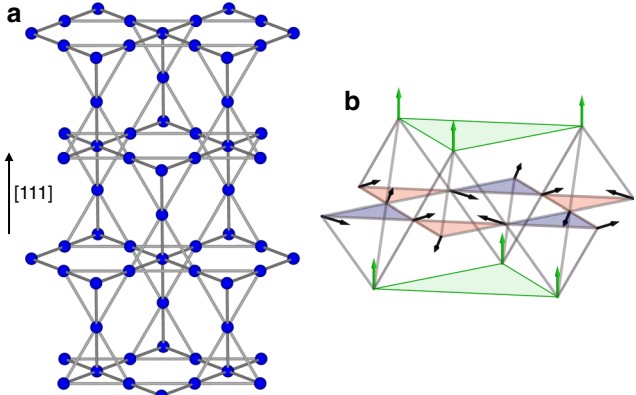

**Fig. 1** Kagome ice state in the pyrochlore lattice. **a** View of the pyrochlore structure along the [111] direction showing the stacking of triangular and kagome planes (see also Supplementary Table 1 and Supplementary Figure 1). **b** Classical kagome ice state when a magnetic field is applied along [111]. The apical spins in the triangular planes (green triangles) are parallel to the field, while the kagome spins obey the ice-rule 2 in $-1$ out (red triangles)/1 in $-2$ out (blue triangles)

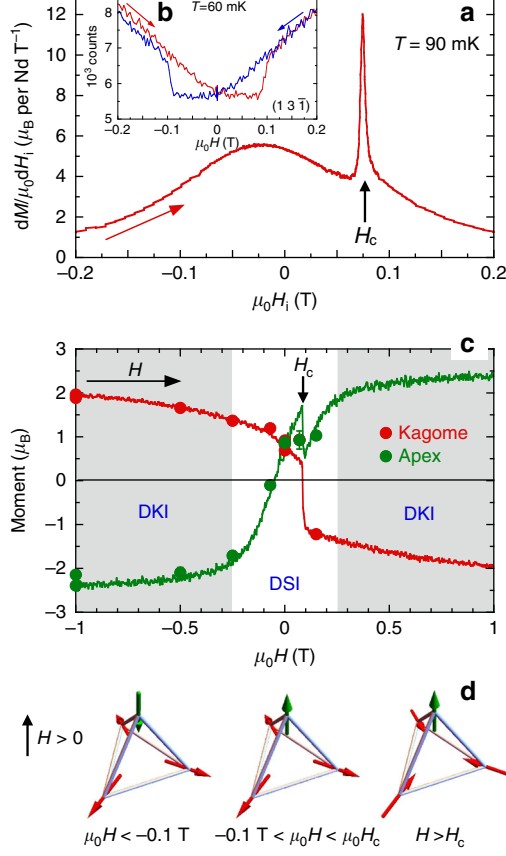

**Fig. 2** Field evolution of the magnetic structure for $H \parallel [111]$. **a** Derivative of the magnetization $dM/dH_i$ vs the internal magnetic field $H_i$ when $H$ is swept from negative to positive. **b** Field dependence of the $(13\bar{1})$ Bragg peak intensity when sweeping the field from negative to positive (red) and back (blue). **c** Refined magnetic moment for the apex (green) and kagome (red) spins of a tetrahedron as a function of field. Points correspond to data collections. The error bars are provided by the Rietveld refinement made using Fullprof. Lines are obtained from the analysis of the field sweeping measurements where the moment values have been rescaled to the refined values obtained from the data (see Supplementary Figures 2–4 and Supplementary Note 1). The obtained moments are aligned along the $\langle 111 \rangle$ directions. By convention, for an up tetrahedron, a positive (negative) moment value corresponds to an out (in) moment. The regions where a dynamic kagome ice state (DKI) is stabilized are colored in gray. At low field, the dynamic spin ice state (DSI) is observed. **d** Schematic of the magnetic structure of the three observed configurations for an up tetrahedron: (i) 1 in −3 out for $\mu_0 H < -0.1$ T, (ii) all out around $H = 0$, (iii) 3 in −1 out for $H > H_c$. Note that only the spin directions are meaningful and not their size: apart from at saturation and at $H = 0$, apex and kagome spins do not carry the same ordered moment

field, the intensity of the arms decreases, but clearly persists up to at least 1 T (see Fig. 3c).

The obtained pattern actually resembles the kagome ice neutron scattering function. Nevertheless, the pinch points expected at $\mathbf{q} = (2/3, 2/3, -4/3)$ (and related symmetry positions), and characteristic of the existence of algebraic correlations, are not clearly defined. This is partly due to the energy integration which tends to broaden the observed features, but also to the nature of the spectrum itself as discussed below. The excitations are broad both in $\mathbf{q}$-space and in energy (see Fig. 4), which tends to smear out the kagome ice features. At the same time, new dispersive branches form. They stem from the positions of the kagome ice pinch points, spread in reciprocal

space and finally close up at about 0.12 meV at $\mathbf{q} = (2, 2, 0)$ (see Fig. 4c, Supplementary Note 2 and Supplementary Figures 5–10). This set of excitations (kagome ice mode and new dispersive branches) can thus be associated with two-dimensional dynamics of the kagome spins. The change towards this two-dimensional regime occurs progressively, as can be seen from the smooth evolution of the spectra. At 0.75 T, a high energy flat mode at about 0.3 meV stands out from these low-energy excitations. This mode can be attributed to the local excitations of the apical spins, which are strongly polarized by the applied field and are not involved in the two-dimensional dynamics. The energy of this mode is thus related to the Zeeman splitting associated to the apical spins.

These observations are consistent with diffraction results shown in Fig. 2c. They reveal that the appearance of the kagome ice pattern on the 70 μeV flat mode, of the new dispersive branches and of the high energy branch matches with the full polarization of the apex spins at $\mu_0 H \geq 0.25$ T. Beyond this value, the dimensional reduction driven by the magnetic field confines the fluctuations to the kagome planes, thus transforming the dynamic spin ice mode into a dynamic kagome ice mode.

## Discussion

Theoretically, owing to the dipolar octupolar nature of the $Nd^{3+}$ electronic ground state, the physics of $Nd_2Zr_2O_7$ can be described by an XYZ Hamiltonian[22] written in the local frame of the effective pseudo-spins 1/2, $\boldsymbol{\tau} = (\tau^x, \tau^y, \tau^z)$, residing on each site of the pyrochlore lattice:

$$\mathcal{H} = \sum_{\langle i,j \rangle} \left[ J_x \tau_i^x \tau_j^x + J_y \tau_i^y \tau_j^y + J_z \tau_i^z \tau_j^z \right. $$
$$\left. + J_{xz} \left( \tau_i^x \tau_j^z + \tau_i^z \tau_j^x \right) \right] + g_z \mu_B \sum_i \mathbf{H} . \mathbf{z}_i \tau_i^z \tag{1}$$

$J_x$, $J_y$, $J_z$, and $J_{xz}$ are effective interactions and $g_z = 4.55$ is the effective g-factor, deduced from the crystal electric field scheme[23] ($g_x = g_y = 0$). $\tau^z$ (along $\langle 111 \rangle$) identifies with the dipolar magnetic moment $S^z = g_z \tau^z$. $\tau^{x,y}$ components are non-observable quantities, which respectively transform as dipolar and octupolar moments under symmetries.

The main difficulty in describing the zero-field ground state of $Nd_2Zr_2O_7$, is to understand the coexistence of an all in–all out ground state characterized by a reduced moment, with a dynamic spin ice mode. Recently, a plausible scenario has been proposed, assuming that the pseudo-spins $\tau$ order in a direction tilted away from their local $z$ magnetic direction, within the $(x, z)$ plane[25]. This state projects onto the $z$ axes as a classical all in–all out configuration, but with a reduced moment. The obtained spin excitation spectrum encompasses a dynamic spin ice mode along with dispersive branches, as observed in the experiment. We have studied the evolution of such a state when a magnetic field is applied along [111]. We find that the apical spins are polarized and the model predicts the appearance of the kagome ice pattern (see Fig. 3b), as well as of the new dispersion stemming from the kagome ice pinch points (see Fig. 4). We have analyzed the spectra measured as a function of field and we have found that, to get the best agreement between our data and the model, the set of exchange parameters given in ref. [25] has to be slightly modified. This new analysis gives a revised set of parameters: $J_x = (-0.36 \pm 0.16)$ K, $J_y = (0.066 \pm 0.2)$ K, $J_z = (0.86 \pm 0.15)$ K, and $J_{xz} = (0.44 \pm 0.15)$ K (see Supplementary Notes 3 and 4 and Supplementary Figures 11–14).

As pointed out in ref. [25], the spin excitation spectrum can be understood by considering the fluctuations of the field emerging from the dynamic components, thus generalizing to the dynamics the concept of emergent field introduced in spin ice[26] (see

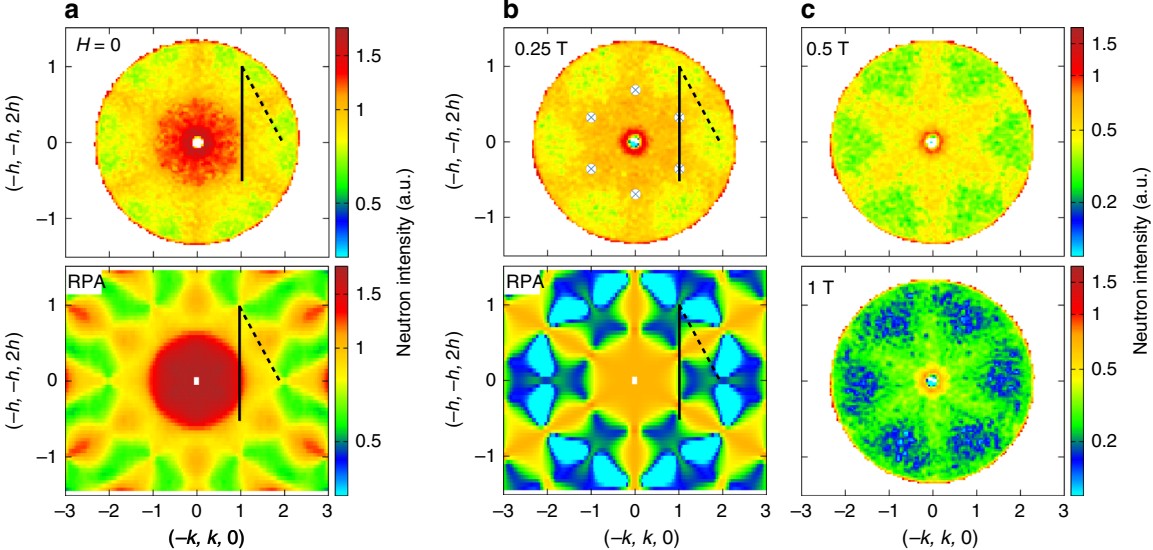

**Fig. 3** Dynamic kagome ice state seen in inelastic neutron scattering for $H \parallel [111]$. Inelastic intensity averaged around $E = (50 \pm 5)$ μeV (corresponding to the energy shown by the red arrow in Fig. 4). **a** Zero-field measurements at 60 mK and Random Field Approximation (RPA) calculations for the pseudo-spin 1/2 model described in the text with $J_x = -0.36$ K, $J_y = 0.066$ K, $J_z = 0.86$ K, and $J_{xz} = 0.44$ K at $T = 0$. **b** Measurements in 0.25 T at 60 mK and calculations with the same parameters. The crossed circles mark the position of the expected pinch points, which appear blurred in the measurements. **c** Measurements at 0.5 and 1 T. The black full and dashed lines indicate the directions of the slices along $(-1-h, 1-h, 2h)$ and $(-2, h, 2-h)$ shown in Fig. 4

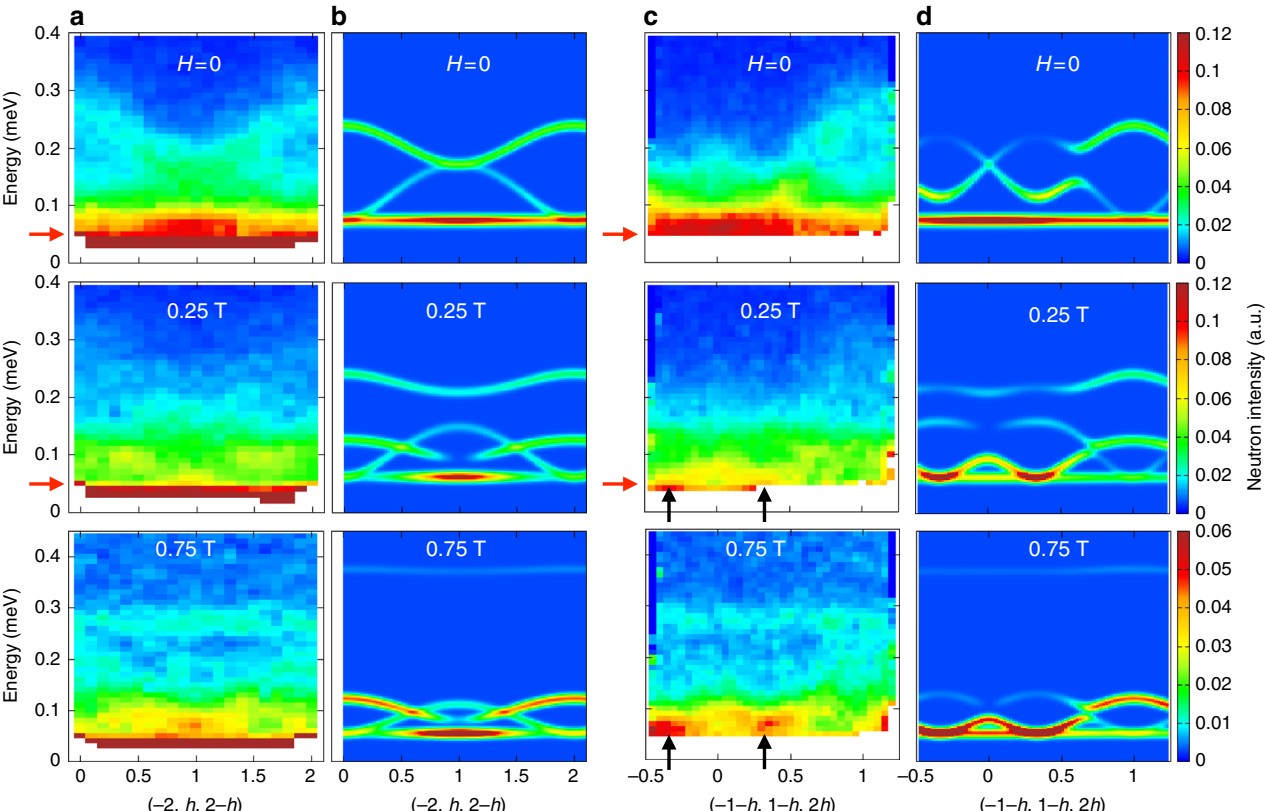

**Fig. 4** Magnetic excitation spectra. Slices along $(-2, h, 2-h)$ ( dashed line in Fig. 3) in zero field, 0.25 and 0.75 T: **a** Measurements at 60 mK. **b** RPA calculations with $J_x = -0.36$ K, $J_y = 0.066$ K, $J_z = 0.86$ K, and $J_{xz} = 0.44$ K at $T = 0$. Slices along $(-1-h, 1-h, 2h)$ ( black line in Fig. 3) in zero field, 0.25 and 0.75 T: **c** Measurements at 60 mK. **d** RPA calculations with the same parameters. The black arrows mark the positions of the kagome ice pinch points, from which the dispersive branches emerge. The red arrows mark the position of the energy cuts of Fig. 3, slightly below the actual position of the mode to minimize the integration of the dispersive branch. The color scale at 0.75 T has been changed to emphasize the flat band at 0.26 meV in the measurements and 0.36 meV in the calculations

Supplementary Note 5 and Supplementary Figures 15, 16). Applying a Helmholtz–Hodge decomposition to these fields gives rise to divergence-free and divergence full dynamic fragments, which can be seen as a quantum analog of the magnetic moment fragmentation[25,27]. The divergence-free part identifies with the flat mode. It is spin ice like in zero field and kagome ice like above 0.25 T. The divergence full part lies in the dispersive branches emerging from the pinch points and corresponds to the propagation of charged quasi-particles. Above 0.25 T, the dimensional reduction confines them to the kagome planes.

The observation of the two-dimensional kagome ice mode over a large field range (0.25–1 T) demonstrates the robustness of this feature. Interestingly, field-induced transitions in the magnetic structure do not directly affect this low-energy inelastic flat mode, showing that the actual magnetic ordered ground state is well protected from these excitations. In the mean field model presented above, this counterintuitive disconnection is due to the fact that pseudo-spins aligned along the $z$ axes do not give rise to transverse spin excitations visible in neutron scattering. In other words, the observable spectrum originates from fluctuations out of the $(x, y)$ pseudo-spin ordered components, having a non-zero projection onto the magnetic $z$ axes. Remarkably, the resulting flat mode remains at the same energy of about 70 μeV, in the whole field range from zero to high field, even though its structure factor, and so the nature of the fluctuations are impacted by the field.

While this XYZ model allows us to describe the main features of our observations, it fails in several aspects, which may call for more sophisticated approaches. First, it cannot account for the field-induced transition at 0.08 T, possibly due to the high value obtained for the $J_x$ parameter, which constrains the pseudo-spins along the local $x$ axis. Second, when the field is increased, the energy of the high-energy flat mode corresponding to flipping of the apical spin is shifted in the model towards higher energy than what is observed (see Fig. 3). Finally, it predicts well-defined pinch points and excitations, while the experimental features appear much broader than the experimental resolution (<20 μeV). This is especially true for $H = 0$ in the vicinity of $\mathbf{q} = (-2, 1, 1)$ and for $H \neq 0$ in the whole $\mathbf{q}$ range, where the spectrum resembles a continuum. The broadness of these features is reminiscent of observations in some other pyrochlore frustrated magnets, where the spin wave excitations are also not well defined. A prominent example is $Yb_2Ti_2O_7$, in which broad spin waves (in some $\mathbf{q}$ regions) emerge from a continuum when a magnetic field is applied, becoming well defined at large field only[28]. These results draw attention to the limits of the conventional spin wave approach in frustrated systems in the presence of quantum effects, which are prone to exotic excitations mediated by complex processes. In the particular case of $Nd_2Zr_2O_7$, the broadness of the spin waves is likely due to interactions between the divergence free and full dynamical fragments. While decoupled at the mean field level of a spin wave approach, we anticipate that they become coupled in more elaborate treatments.

Finally, recent theoretical studies of the XYZ Hamiltonian in the presence of a [111] magnetic field discussed above have proposed the existence of a quantum kagome ice phase[13,14]. The specific exchange parameters obtained for $Nd_2Zr_2O_7$ locate it quite far from this phase, but the observation of a kagome ice mode in the excitation spectrum paves the way for further explorations. We anticipate that rich physics, from both a theoretical and an experimental point of view, has still to be discovered in this system.

## Methods

**Synthesis**. Single crystals of $Nd_2Zr_2O_7$ were grown by the floating-zone technique using a four-mirror xenon arc lamp optical image furnace[29,30].

**Inelastic neutron scattering**. Inelastic neutron scattering experiments were carried out at the Institute Laue Langevin (ILL, France) on the IN5 disk chopper time of flight spectrometer and operated with $\lambda = 8.5$ Å. The $Nd_2Zr_2O_7$ single crystal sample was attached to the cold finger of a dilution insert and the field was applied along [111]. The sample was misaligned by about 2° around the $(1, -1, 0)$ axis and no misalignment could be detected around the $(-1, -1, 2)$ axis. The data were processed with the Horace software, transforming the recorded time of flight, sample rotation, and scattering angle into energy transfer and $\mathbf{q}$-wave-vectors. Maps were symmetrized to account for the 6-fold symmetry in this scattering plane. It is worth noting that the magnetic structure factor is not favorable in the scattering plane of these experiments, compared to the case of scattering plane perpendicular to the $[1\bar{1}0]$ direction.

**Neutron diffraction**. The neutron diffraction data were taken at the D23 single crystal diffractometer (CEA-CRG, ILL France) using a copper monochromator and $\lambda = 1.28$ Å. Here, the field was applied along the [111] direction. Refinements were carried out with the Fullprof software suite[31] on data collections of 60 Bragg peaks. The sample was first saturated in $-3$ T, and data collections were made by increasing the field step by step towards the measurement value. In addition, measurements were performed by collecting the intensity on the top of some Bragg peaks when sweeping the magnetic field from $-1$ to 1 T at a rate of 14.8 mT min$^{-1}$.

**Magnetization**. Magnetization measurements were performed on a SQUID magnetometer equipped with a dilution refrigerator[32] on a parallelepiped sample[23]. The field was swept from $-0.4$ to 0.4 T. The magnetization is corrected for demagnetization effects.

**Calculations**. Calculations were carried out on the basis of a mean field treatment of the XYZ Hamiltonian written in terms of a pseudo-spin 1/2 spanning the CEF doublet ground state. The spin dynamics were then calculated numerically in the Random Phase Approximation[33–36].

## Data availability

All relevant data are available from the authors. Inelastic neutron scattering data performed at the ILL are available at 10.5291/ILL-DATA.4-05-637.

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

## Acknowledgements

The authors acknowledge P.C.W. Holdsworth for useful discussions and C. Paulsen for the use of his dilution SQUID magnetometer. E.L. acknowledges financial support from ANR, France, Grant No. ANR-15-CE30-0004. The work at the University of Warwick was supported by the Engineering and Physical Sciences Research Council (EPSRC), United Kingdom, through Grant No. EP/M028771/1.

## Author contributions

Crystal growth and characterization were performed by M.C.H., M.R.L., and G.B. Inelastic neutron scattering experiments were carried out by S.P., E.L., J.O., and H.M. Diffraction experiments were carried out by S.P., E.L., and E.R. Magnetization measurements were performed by E.L. The data were analyzed by S.P., E.L. with input from E.R., and J.O. RPA calculations were carried out by S.P. The paper was written by E.L. and S.P. with feedback from all authors.

## Additional information

**Competing interests:** The authors declare no competing interests.

