## [Peer Review File · Nature Communications]

Reviewers' comments:

Reviewer #1 (Remarks to the Author):

Dear editor:

I have had the opportunity to review the submission, "Dynamic quantum kagome ice," by Lhotel et al. This work expands upon previous neutron scattering measurements on Nd₂Zr₂O₇ in applied magnetic fields along the <111> direction to stabilize dynamic kagome ice states.

The main results of the paper include (i) the smearing of pinch points in the kagome ice state as revealed by neutron diffraction and (ii) the breakdown of spin waves within the kagome state. Both of these results are important in establishing the unconventional spin dynamics in applied fields. The main conclusions within this paper, if true, would be an important contribution towards the field of geometrically frustrated magnetism. It would also be of broad interest to the physics community. However, I have some concerns that should be addressed before considering this work for publication:

(1) The diffuse scattering profiles in figure 3 show the smearing of pinch points in applied fields. However, it is important to stress that the contributions are not completely elastic - there is an intrinsic energy width. It is a bit misleading to claim that these pinch point excitations are smeared solely because of the dynamic kagome state. The sample also appears to be slightly misaligned (seen mostly clearly in the 1 T data), since the diffuse scattering data is not symmetric around $(-k, k, 0)$.

(2) The authors claim that the breakdown of the inelastic modes in an applied field are due to unconventional spin dynamics (figure 4). A more plausible explanation could be that the lack of agreement between the sharp modes calculated and the broad modes observed could be similar to what is observed in Yb₂Ti₂O₇ in applied fields by Thompson et al (PRL 2017). In Yb₂Ti₂O₇, which has been touted as a quantum spin ice, the sharp modes predicted by spin wave theory become smeared due to quasiparticle breakdown. While the energy scales are different for the two states (as evidenced by the different J constants), the breakdown of classical spin waves is seen for both. As an aside, the breakdown of spin waves has also been seen in Er₂Pt₂O₇ (Hallas et al, PRL 2017), with completely different physics, so this is not a unique feature of the 'dynamic kagome state.' The physics is different for the three different systems, but similar features are seen in the spin wave spectra. Therefore, the breakdown of spin excitations in

applied fields for Nd₂Zr₂O₇ is not unique, and is not a "smoking gun" for the dynamic kagome state. I agree that the smearing is likely due to dominant quantum exchange terms in the Hamiltonian. However, this is not "new" and is now a more well established phenomena in geometrically frustrated quantum magnets.

(3) As the authors point out, there are other significant issues with the spin wave calculations. The field induced transition at $H = 0.08$ T cannot be accounted for. The high energy mode is shifted to higher fields than the model. And the lack of pinch points in the data is not accounted for in the model. The latter point is easier to explain. However, the first two points should be elucidated in more detail before publication.

Reviewer #2 (Remarks to the Author):

This paper reports on magnetization and neutron scattering measurements on Nd₂Zr₂O₇ in a [111] applied magnetic field, and associated theoretical modeling, finding evidence for a dynamical quantum kagome ice mode in this system. If my understanding is correct, this is a gapped and nearly dispersionless mode that is an excitation above the 3-in 1-out / 3-out 1-in ground state. The dispersionless nature comes from the fact that, to a reasonable approximation, if one decomposes spin configurations into "longitudinal" and "transverse" parts, the energy does not depend on the transverse part. The interpretation of the data looks very solid to me. Moreover, the manuscript is clear and well written.

I think these results are very interesting, but I do not see how they open the door to study of the quantum kagome ice state recently studied theoretically, as the authors claim. That state exists in an apparently very different parameter regime, where the kagome planes are in a 2-in 1-out / 2-out 1-in configuration in the ground state. While the flat mode apparently has some character of the quantum kagome ice ground state, these are not at all the same thing -- in particular, the flat mode is a single excitation above a ground state that is still mostly 3-in 1-out / 3-out 1-in. If the authors want to make their claim about relationship to the quantum kagome ice ground state, they should give more details what they have in mind.

Nonetheless, even without this claim, I recommend publication in Nature Communications. For one thing, these results are an interesting example of how a spin system with an apparently simple classical ground state can still have very non-trivial geometrical features in its excitation spectrum.

Reviewer #3 (Remarks to the Author):

Exquisitely energy resolved inelastic neutron scattering is used to determine the $B_{||}(111)$ dependence of spin excitation in $\text{Nd}_2\text{Zr}_2\text{O}_7$. Quality of experiment is quite impressive.

Authors argue that the spectra in Fig 4 along cuts in Fig 3 support the expected picture of in-plane kagome spin ice and its spin dynamics. The dispersive branches emerging from KSI pinch points are key evidence.

At this level im inclined to accept the presentation. But actual examination of the raw data in Fig 4B show that is is not really of sufficiently high SNR to be certain. Moreover the discussion in the final paragraph of excitations disconnected from the broken symmetry ordered state makes little sense in tradition spin dynamics.

I encourage the authors to edit ms making clearer the quantitative confidence of their claim of detecting dynamics of quantum kagome ice. This may benefit them later if a different possibel explanation emerges 0 that often happens.

Reply to Reviewers

Reviewer 1

We thank referee 1 for his/her careful reading of the manuscript, the positive assessment of the quality and importance of our work. We have hopefully taken all their comments into account as detailed below and have modified the manuscript accordingly.

Dear editor:

I have had the opportunity to review the submission, “Dynamic quantum kagome ice,” by Lhotel et al. This work expands upon previous neutron scattering measurements on Nd₂Zr₂O₇ in applied magnetic fields along the $\langle 111 \rangle$ direction to stabilize dynamic kagome ice states.

The main results of the paper include (i) the smearing of pinch points in the kagome ice state as revealed by neutron diffraction and (ii) the breakdown of spin waves within the kagome state. Both of these results are important in establishing the unconventional spin dynamics in applied fields. The main conclusions within this paper, if true, would be an important contribution towards the field of geometrically frustrated magnetism. It would also be of broad interest to the physics community. However, I have some concerns that should be addressed before considering this work for publication:

(1) The diffuse scattering profiles in figure 3 show the smearing of pinch points in applied fields. However, it is important to stress that the contributions are not completely elastic - there is an intrinsic energy width. It is a bit misleading to claim that these pinch point excitations are smeared solely because of the dynamic kagome state.

Answer: The dynamic kagome ice mode is indeed observed at finite energy, and is consequently inelastic in nature. Nevertheless, as stressed by the referee, it has a finite energy width, in the same way as the dynamic spin ice mode observed in zero field.

Actually, in zero field, the pinch points of the spin ice mode are quite well defined (See *Nat. Phys.* **12**, 746-750, 2016), but they are not present in the scattering plane of Figure 3. In the field induced regime, the pinch points of the kagome ice mode are smeared, and not well defined, contrary to what is obtained in the calculations performed in the XYZ model. Note that we have not claimed in the manuscript that “the pinch point excitations are smeared solely because of the dynamic kagome state”. We feel important to clarify this point: as seen in Figure 4, all the features of the experimental spectrum are very broad and the kagome ice mode itself does not escape this trend. This is certainly due to complex phenomena that we discuss in the response to the point (2). In addition, dispersive branches emerge from the positions of the kagome ice pinch points, so that when the intensity is averaged over the 45-55 μeV range to obtain the mode shown in Figure 3 (b) and (c), the smearing is reinforced.

We have reworded this point in the manuscript.

FIG. 1. Elastic cut of the scattering in reciprocal space. The integration range is $[-50, 50] \mu\text{eV}$.

The sample also appears to be slightly misaligned (seen mostly clearly in the 1 T data), since the diffuse scattering data is not symmetric around $(-k, k, 0)$.

Answer: The diffuse scattering profiles have been symmetrized, as indicated in the Methods section. The asymmetry noted by the referee on one branch at 1 T is thus an artifact. Figure 1 displays a zero energy cut of the scattering, allowing one to directly observe the intensities of the (220) (and symmetry related) Bragg peaks. In principle, if the sample was perfectly aligned, they should be identical. Examination of their relative intensities thus gives an upper bound of the misalignment. It is estimated to ± 2 degrees around the $(-1, 1, 0)$ axis and 0 degrees around the $(-1, -1, 2)$ axis. Within the context of our study, this misalignment does not affect the physical conclusions arrived at in this manuscript, i.e. the existence of a field induced kagome ice mode in $\text{Nd}_2\text{Zr}_2\text{O}_7$.

This information is now reported in the Methods section.

(2) The authors claim that the breakdown of the inelastic modes in an applied field are due to unconventional spin dynamics (figure 4). A more plausible explanation could be that the lack of agreement between the sharp modes calculated and the broad modes observed could be similar to what is observed in $\text{Yb}_2\text{Ti}_2\text{O}_7$ in applied fields by Thompson et al (PRL 2017). In $\text{Yb}_2\text{Ti}_2\text{O}_7$, which has been touted as a quantum spin ice, the sharp modes predicted by spin wave theory become smeared due to quasiparticle breakdown. While the energy scales are different for the two states (as evidenced by the different J constants), the breakdown of classical spin waves is seen for both. As an aside, the breakdown of spin waves has also been seen in $\text{Er}_2\text{Pt}_2\text{O}_7$ (Hallas et al, PRL 2017), with completely different physics, so this is not a unique feature of the 'dynamic kagome state.' The physics is different for the three different systems, but similar features are seen in the spin wave spectra. Therefore, the breakdown of spin excitations in applied fields for $\text{Nd}_2\text{Zr}_2\text{O}_7$ is not unique, and is not a "smoking gun" for the dynamic kagome state. I agree that the smearing is

likely due to dominant quantum exchange terms in the Hamiltonian. However, this is not “new” and is now a more well established phenomena in geometrically frustrated quantum magnets.

Answer: We agree that the breakdown of classical spin waves has been seen in other frustrated systems. For instance, $\text{Yb}_2\text{Ti}_2\text{O}_7$ is an extreme example, where the classical spin wave picture simply does not apply, which clearly features unconventional spin dynamics. In less extreme cases, one expects the formation of a complex response encompassing a weak two particle continuum superimposed on a broadened single-quasiparticle dispersion. This results in modified spin dynamics compared to the conventional case, and $\text{Nd}_2\text{Zr}_2\text{O}_7$ falls into this class. To avoid any confusion in the manuscript, we have replaced the word “unconventional” by “exotic”.

Note that we have not argued in the manuscript that such “exotic” spin dynamics is a fingerprint of the dynamic kagome state. The experimental evidence for the existence of the dynamic kagome ice state is the presence of a flat mode, with a six-arm pattern in the neutron scattering function.

We should also point out that even if the breakdown of spin wave excitations is indeed well established from an experimental point of view in some frustrated pyrochlore systems, to our knowledge, no theoretical model has been published to reproduce such observations, so that the mechanisms at play are not elucidated yet. In that sense, the general character of such unusual inelastic spectra, that are also observed here in $\text{Nd}_2\text{Zr}_2\text{O}_7$, is an important point for the general understanding of the geometrically frustrated systems in the presence of quantum effects.

To extend the discussion, we would like however to point to strong differences between $\text{Nd}_2\text{Zr}_2\text{O}_7$ and the two systems mentioned by the referee, $\text{Yb}_2\text{Ti}_2\text{O}_7$ and $\text{Er}_2\text{Pt}_2\text{O}_7$. First, it is important to note that the nature of the magnetic moment itself is very different, since the Nd ion has an Ising dipolar octupolar doublet, while Er and Yb ions have an XY anisotropy, which considerably modifies the spin dynamics.

Second, in $\text{Er}_2\text{Pt}_2\text{O}_7$, only one study in zero field on a powder sample has been reported which provided no evidence of spin waves, maybe due to the proximity to competing phases. An anisotropy gap is present in the ordered state, as predicted theoretically for the Palmer-Chalker phase. It should be noted that a study of the related compound $\text{Er}_2\text{Sn}_2\text{O}_7$ shows well defined spin waves in the field polarized phase, even on a powder sample.

Third, in $\text{Yb}_2\text{Ti}_2\text{O}_7$, for which more experimental studies are available, and where the role of several competing phases has also been evoked, well defined spin waves are only observed in a high magnetic field, together with the formation of a two magnon continuum. The zero field spectrum does not present well defined excitations at all.

In $\text{Nd}_2\text{Zr}_2\text{O}_7$, the flat mode and the associated spin waves are rather “well-defined” in the whole field range: although the features are broader than usual spin wave excitations (a weak continuum of excitations likely exists to some extent), the neutron data clearly feature a dispersion, similar in the \mathbf{q} -space as the model predictions, suggesting that the concept of spin wave quasi-particle is still valid. Some complex processes involving scattering of several quasi-particles are likely at play in the spin dynamics, but do not affect the main picture of a kagome ice mode.

Following the referee’s remark, we have extended the discussion to mention some similarities with other pyrochlore frustrated magnets, but without going into too many details, to avoid straying from the main message of the manuscript.

(3) As the authors point out, there are other significant issues with the spin wave calculations. The field induced transition at $H = 0.08$ T cannot be accounted for. The high energy mode is shifted to higher fields than the model. And the lack of pinch points in the data is not accounted for in the model. The latter point is easier to explain. However, the first two points should be elucidated in more detail before publication.

Answer: as pointed out by the referee (and written explicitly in the manuscript), the mean field and spin wave approach to the XYZ Hamiltonian cannot account for the value of the metamagnetic field, the field dependence of the high energy mode, as well as the blurring of the pinch points, and more generally the width of the observed inelastic features. We have of course tried other analysis to account for these observations, especially the first two points.

At the moment, the only *heuristic* way we have found to reproduce these features is by considering two fragments, as initially proposed in the fragmentation picture. Those fragments behave independently, and are characterized by different g -factors. The first one displays the all in – all out ordering and is associated with an antiferromagnetic J_z . Importantly, since the moments of this fragment are aligned along the $\langle 111 \rangle$ axes, it has no dynamics. The second fragment is more complex and is in principle described by an XYZ Hamiltonian. We find that to reproduce the inelastic neutron scattering spectra, its ground state, at the mean field level, is a uniform octupolar ordering where all the pseudo spins point along the y octupolar local axis. From this peculiar state emerge a spin wave spectrum consisting in a flat spin ice mode and dispersing branches. Under an applied $[111]$ field, the all in – all out part flips, as expected, towards a 3 in – 1 out or 3 out – 1 in state. In contrast, the second fragment is continuously polarized by the field. Furthermore, the calculated spin dynamics under these assumptions is very similar to the calculations presented here, except that the high energy mode is less shifted. The problem, however, is to reconcile these two fragments to obtain a single Hamiltonian able to describe both degrees of freedom and thus the whole system. We have not succeeded in solving this issue at the moment.

This approach may nevertheless find some justification in a “quantum” variant of the fragmentation phenomenon. Here, the divergence full part would identify with the all in – all out fragment, like in the “classical” magnetic fragmentation. But the divergence free part would then consist in a quantum spin liquid which would encompass the other part of the magnetic moment. This quantum spin liquid would not exhibit the elastic spin ice pattern expected for a classical Coulomb phase, and may have gapped excitations. Applying the field along $[111]$ would then result in modifying the nature of the quantum liquid state and may lead to phenomena with a

kagome ice mode similar to what we have observed. At this stage, such a picture appears speculative, and sophisticated theoretical tools have to be developed, in order to validate this approach.

We thus believe that these analyses go far beyond the scope of this manuscript, which first aims at evidencing experimentally the existence of a dynamic kagome ice state and to compare the experimental results with the “simple” theoretical available picture proposed by O. Benton to describe $\text{Nd}_2\text{Zr}_2\text{O}_7$: in this sense, we show that although this theory gives a qualitative description of our observations, it fails in several points (also raised by the referee) which may be crucial (or not) for the understanding of the system. For these reasons, we are convinced that our study will arouse new theoretical studies, maybe starting from different approaches, which will help having a global picture of the nature of the partial ordered state, and field induced states of $\text{Nd}_2\text{Zr}_2\text{O}_7$.

Reviewer 2

We gratefully thank Reviewer 2 for the interest he/she shows in our results and for recommending our work for publication.

This paper reports on magnetization and neutron scattering measurements on $\text{Nd}_2\text{Zr}_2\text{O}_7$ in a [111] applied magnetic field, and associated theoretical modeling, finding evidence for a dynamical quantum kagome ice mode in this system. If my understanding is correct, this is a gapped and nearly dispersionless mode that is an excitation above the 3-in 1-out / 3-out 1-in ground state. The dispersionless nature comes from the fact that, to a reasonable approximation, if one decomposes spin configurations into “longitudinal” and “transverse” parts, the energy does not depend on the transverse part. The interpretation of the data looks very solid to me. Moreover, the manuscript is clear and well written.

I think these results are very interesting, but I do not see how they open the door to study of the quantum kagome ice state recently studied theoretically, as the authors claim. That state exists in an apparently very different parameter regime, where the kagome planes are in a 2-in 1-out / 2-out 1-in configuration in the ground state. While the flat mode apparently has some character of the quantum kagome ice ground state, these are not at all the same thing – in particular, the flat mode is a single excitation above a ground state that is still mostly 3-in 1-out / 3-out 1-in. If the authors want to make their claim about relationship to the quantum kagome ice ground state, they should give more details what they have in mind.

Nonetheless, even without this claim, I recommend publication in Nature Communications. For one thing, these results are an interesting example of how a spin system with an apparently simple classical ground state can still have very non-trivial geometrical features in its excitation spectrum.

Answer: We have clarified the discussion about the relation between the quantum kagome ice ground state and our observations, and we explain why we mentioned this state below.

The important point in our results is that the kagome flat mode seems to be associated with a

two dimensional excitation of the kagome planes in a 3 in / 3 out configuration, which would obey in a sense the 2-in 1-out / 2-out 1-in kagome ice rule. The apical spins are “integrated” out by the field, giving rise to another excitation at higher energy, whose position moves with the magnetic field.

Theoretical work on the XYZ Hamiltonian under a [111] applied field has shown the existence of a quantum kagome ice ground state. We have referred to this work precisely because it is the same Hamiltonian as the one proposed to describe $\text{Nd}_2\text{Zr}_2\text{O}_7$. Of course, as noted by the referee, the J parameters in our case do not stabilize the quantum kagome ice state. But it appeared interesting to us to make the connection between the observation of a kagome ice mode in the excitations and the quantum kagome ice state found in another part of the XYZ phase diagram. Experimentally one could imagine to try and tune the parameters of $\text{Nd}_2\text{Zr}_2\text{O}_7$ (by applying pressure or chemical doping) to get closer to the predicted quantum kagome ice phase.

Reviewer 3

We finally would like to thank referee 3 for the positive assessment of the quality and importance of our work.

Exquisitely energy resolved inelastic neutron scattering is used to determine the $B \parallel (111)$ dependence of spin excitation in $\text{Nd}_2\text{Zr}_2\text{O}_7$. Quality of experiment is quite impressive.

Authors argue that the spectra in Fig 4 along cuts in Fig 3 support the expected picture of in-plane kagome spin ice and its spin dynamics. The dispersive branches emerging from KSI pinch points are key evidence.

At this level im inclined to accept the presentation. But actual examination of the raw data in Fig 4B show that is is not really of sufficiently high SNR to be certain.

Answer: Although the signal to noise ratio is less good in the present scattering plane than for instance in the scattering plane perpendicular to the $[1\bar{1}0]$ direction, the main features we observe at finite energy clearly emerge above the experimental noise. The comparison between the experimental results of Figure 4 and the calculations show several important points: the \mathbf{q} -values where the highest intensity is observed are consistent with the spin-wave positions in the calculations. As discussed above, the features are however much broader than the calculations, and a continuous signal is present above the flat mode, contrary to calculations. This continuous signal is clearly well above the experimental noise as can be seen in energy cuts, except in the 0.75 T case (bottom of the figure 4) where it is a bit less defined because the signal to noise ratio is smaller and the intensity color scale has been modified to enhance the observed features.

Moreover the discussion in the final paragraph of excitations disconnected from the broken symmetry ordered state makes little sense in tradition spin dynamics.

Answer: We agree with the referee, and indeed, it is one of the reasons which makes our results exciting ! We believe that it is the specific nature of the Nd ion, with its dipolar-octupolar doublet which induces this “disconnection” between the ordered state and the excitations. Let us recall that the z component of the pseudo spin only identifies with the magnetic moment. As a result, a conventional ordered structure with the spins along the z axes does not exhibit visible transverse spin excitations at all. This is the case here for the observed partial all in - all out ordered state. This explains why the spin excitations are apparently independent of the ordered configuration observed by diffraction measurements.

I encourage the authors to edit ms making clearer the quantitative confidence of their claim of detecting dynamics of quantum kagome ice. This may benefit them later if a different possible explanation emerges 0 that often happens.

Answer: We thank the referee for this advice. We believe that the modified manuscript is clear in terms of the interpretations we provide for our observed data and we have also made an attempt to leave it open to alternative explanations.

Reviewers' comments:

Reviewer #1 (Remarks to the Author):

Dear editor

I have recently reviewed the revised manuscript and the referee comments. While I think that certain aspects of the manuscript have been improved, I also feel that other aspects leave more unanswered questions. I would be leaning towards rejecting this paper if these questions are not resolved in future drafts.

Major points:

(1) In response to my initial concerns about the inability of the authors to fit the spin wave data, the following response was given:

"We agree that the breakdown of classical spin waves has been seen in other frustrated systems. For instance, $\text{Yb}_2\text{Ti}_2\text{O}_7$ is an extreme example, where the classical spin wave picture simply does not apply, which clearly features unconventional spin dynamics."

This is not completely accurate. For some regions of the applied field phase diagram, spin wave theory works beautifully (see R. Coldea's work). In other areas it does not. The authors continue with:

"In less extreme cases, one expects the formation of a complex response encompassing a weak two particle continuum superimposed on a broadened single-quasiparticle dispersion. This results in modified spin dynamics compared to the conventional case, and $\text{Nd}_2\text{Zr}_2\text{O}_7$ falls into this class. To avoid any confusion in the manuscript, we have replaced the word "unconventional" by "exotic"."

It is fine to replace the description of the spin waves with a different adjective. It does not take away from the fact that the authors have not adequately described the spin wave spectrum.

The authors continue with:

"Note that we have not argued in the manuscript that such "exotic" spin dynamics is a fingerprint of the dynamic kagome state. The experimental evidence for the existence of the dynamic kagome ice state is the presence of a flat mode, with a six-arm pattern in the neutron scattering function."

I do not believe that this is the single unique signature of a dynamic kagome ice state (especially in the absence of a theory or fitted data, or with a misaligned sample as pointed out by the authors). Do the authors have a reference for this with your estimated exchange parameters? On page 9 of your paper the authors state:

"Finally, recent theoretical studies of the XYZ Hamiltonian in the presence of a [111] magnetic field discussed above have proposed the existence of a quantum kagome ice phase^{13,14}. The specific exchange parameters obtained for Nd₂Zr₂O₇ locate it quite far from this phase, but the observation of a kagome ice mode in the excitation spectrum paves the way for further exploration."

The exchange parameters for this phase are not consistent with Nd₂Zr₂O₇. How do you explain this?

The authors continue their response with:

"We should also point out that even if the breakdown of spin wave excitations is indeed well established from an experimental point of view in some frustrated pyrochlore systems, to our knowledge, no theoretical model has been published to reproduce such observations, so that the mechanisms at play are not elucidated yet. In that sense, the general character of such unusual inelastic spectra, that are also observed here in Nd₂Zr₂O₇, is an important point for the general understanding of the geometrically frustrated systems in the presence of quantum effects."

Again, I will agree that there are unusual excitations here. However, extraordinary claims need extraordinary evidence. Without a model that fits your data it is difficult to claim that these excitations are solely from a dynamic kagome state.

The authors continue with:

"To extend the discussion, we would like however to point to strong differences between Nd₂Zr₂O₇ and the two systems mentioned by the referee, Yb₂Ti₂O₇ and Er₂Pt₂O₇. First, it is important to note that the nature of the magnetic moment itself is very different, since the Nd ion has an Ising dipolar octupolar doublet, while Er and Yb ions have an XY anisotropy, which considerably modifies the spin dynamics.

Second, in Er₂Pt₂O₇, only one study in zero field on a powder sample has been reported which provided no evidence of spin waves, maybe due to the proximity to competing phases. An anisotropy gap is present in the ordered state, as predicted theoretically for the Palmer-Chalker phase. It should be noted that a study of the related compound Er₂Sn₂O₇ shows well defined spin waves in the field polarized phase, even on a powder sample.

Third, in $\text{Yb}_2\text{Ti}_2\text{O}_7$, for which more experimental studies are available, and where the role of several competing phases has also been evoked, well defined spin waves are only observed in a high magnetic field, together with the formation of a two magnon continuum. The zero field spectrum does not present well defined excitations at all.

In $\text{Nd}_2\text{Zr}_2\text{O}_7$, the flat mode and the associated spin waves are rather “well-defined” in the whole field range: although the features are broader than usual spin wave excitations (a weak continuum of excitations likely exists to some extent), the neutron data clearly feature a dispersion, similar in the q-space as the model predictions, suggesting that the concept of spin wave quasi-particle is still valid. Some complex processes involving scattering of several quasi-particles are likely at play in the spin dynamics, but do not affect the main picture of a kagome ice mode."

I agree that the physics is different in the other pyrochlore systems. I was just pointing out that quasiparticle breakdown is not a unique phenomena, nor a smoking gun for dynamic kagome ice states.

With respect to the comments about "flat modes" in $\text{Nd}_2\text{Zr}_2\text{O}_7$: the authors point out two different perspectives about these. In one perspective, the modes are due to the lifting of the spin doublet (page 6). Later on the paper (page 9), the authors claim that the excitations are rather robust to applied fields. How can both perspectives be present? If the modes are lifted doublets, does that not indicate that they are affected by applied fields (as they should be?).

In the absence of a model for the data, the paper reports six-fold symmetry inelastic neutron scattering features in applied magnetic fields for $\text{Nd}_2\text{Zr}_2\text{O}_7$. The rest is unfortunately speculation. I agree that it is likely that theory cannot explain the data. This paper could be justified, if the editor sees fit, as a motivation for theorists to tackle the problem. At the moment, though, I find the results messy and difficult to interpret beyond the discovery of a new broadened excitation, which could be due to quasiparticle breakdown or some more exotic physics. There simply isn't enough data or theory to solidify the main thesis of the work.

(2) In my last point of my first review:

As the authors point out, there are other significant issues with the spin wave calculations. The field induced transition at $H = 0.08$ T cannot be accounted for. The high energy mode is shifted to higher fields than the model. And the lack of pinch points in the data is not accounted for in the model. The latter point is easier to explain. However, the first two points should be elucidated in more detail before publication.

The authors respond with:

"Answer: as pointed out by the referee (and written explicitly in the manuscript), the mean field and spin wave approach to the XYZ Hamiltonian cannot account for the value of the metamagnetic field, the field dependence of the high energy mode, as well as the blurring of the pinch points, and more generally the width of the observed inelastic features. We have of course tried other analysis to account for these observations, especially the first two points."

The authors then discuss their approaches (which do not work), and suggest that this paper might stimulate future theoretical work. I am not opposed to this line of reasoning as justification for the publication of this work. However, I do think that Nature publications are, in general, very rigorous and that experimental work submitted for publication should be accompanied by theoretical backing. In many pyrochlores, it is known that spin wave theory breaks down for certain parts of the phase diagrams, and this is what drives many physicists to study these compounds. However, in this case, the experimentalists have a sixfold smeared out inelastic scattering pattern at low temperatures in Nd₂Zr₂O₇ in an applied field. The sixfold symmetry of their pattern is anticipated (or at least not unlikely) from previous work. The smearing is not, but it is not entirely unexpected given that other pyrochlores have this behaviour (for various reasons). What is the new physics here, then? Without the proper J constants, this work cannot be mapped onto what O. Benton and others are modelling for the Nd pyrochlores. How can this be placed on their (rich) phase diagrams? Recent papers by Melko and others are also at odds with this work.

Minor points:

(1) In the last sentence of the abstract, there is a spelling error (manifests) and the sentence is not grammatically correct.

(2) On page 9, there is an awkward statement about "the broadness of the feature reminds observation..." This should be reworded.

Reviewer #3 (Remarks to the Author):

The research is of excellent experimental quality and the ideas presented are highly original. Certainly this paper will influence the direction of the field.

The revised version is much clearer and throws into sharp relief the key observations and claims. Application of field along 111 disconnects the apical spin producing what could be classical kagome ice patterns in plane; there is good experimental evidence for this. A new flat dynamical mode also appears which is attributed to quantum dynamics of the out of plane component of the same spins whose in plane components appear to be near classical kagome ice. This is an

intriguing hypothesis but is not proven definitively.

I support publication of this fascinating paper which is now incisive, compelling and important - even if some points lack iron-clad proof.

Reply to Reviewers

Reviewer 1

We have carefully read Reviewer 1's comments, and we still beg to differ on a number of important points. We present below a point-by-point response to the views expressed by Referee 1.

(1) In response to my initial concerns about the inability of the authors to fit the spin wave data, the following response was given: "We agree that the breakdown of classical spin waves has been seen in other frustrated systems. For instance, $\text{Yb}_2\text{Ti}_2\text{O}_7$ is an extreme example, where the classical spin wave picture simply does not apply, which clearly features unconventional spin dynamics." This is not completely accurate. For some regions of the applied field phase diagram, spin wave theory works beautifully (see R. Coldea's work). In other areas it does not.

We agree with the referee. We were only discussing the zero field excitations of $\text{Yb}_2\text{Ti}_2\text{O}_7$.

The authors continue with:

"In less extreme cases, one expects the formation of a complex response encompassing a weak two particle continuum superimposed on a broadened single-quasiparticle dispersion. This results in modified spin dynamics compared to the conventional case, and $\text{Nd}_2\text{Zr}_2\text{O}_7$ falls into this class. To avoid any confusion in the manuscript, we have replaced the word unconventional by exotic. It is fine to replace the description of the spin waves with a different adjective. It does not take away from the fact that the authors have not adequately described the spin wave spectrum.

We beg to disagree, and challenge the use of the adverb "adequately". Contrary to what the referee writes, we assess that *the main features of the spectra are indeed captured by our model*. Evidence for this is provided by the calculations in zero field which match the data accurately. Moreover, the finite field calculations are also consistent with the experiment.

The authors continue with:

"Note that we have not argued in the manuscript that such exotic spin dynamics is a fingerprint of the dynamic kagome state. The experimental evidence for the existence of the dynamic kagome ice state is the presence of a flat mode, with a six-arm pattern in the neutron scattering function." I do not believe that this is the single unique signature of a dynamic kagome ice state (especially in the absence of a theory or fitted data, or with a misaligned sample as pointed out by the authors). Do the authors have a reference for this with your estimated exchange parameters?

We agree with the referee that this is probably not "the single unique signature of a dynamic kagome ice state", but we have not claimed that this is the case. However, we maintain that this is a necessary condition: a dynamic kagome ice should include a kagome pattern in its excitation spectrum. Next, we cannot provide any reference since it is the first time that this physics is observed experimentally and no theoretical study has predicted it explicitly.

On page 9 of your paper the authors state:

“Finally, recent theoretical studies of the XYZ Hamiltonian in the presence of a [111] magnetic field discussed above have proposed the existence of a quantum kagome ice phase. The specific exchange parameters obtained for Nd₂Zr₂O₇ locate it quite far from this phase, but the observation of a kagome ice mode in the excitation spectrum paves the way for further exploration.”

The exchange parameters for this phase are not consistent with Nd₂Zr₂O₇. How do you explain this?

We sense here that the referee is confusing the *kagome ice mode* that we have measured and the *quantum kagome ice state* discussed by Melko et al., which is indeed different. Actually, the range of parameters considered by R. Melko et al. is very different from those relevant to the case of Nd₂Zr₂O₇. This work addresses another part of the phase diagram (this is written explicitly in the text). More precisely, the quantum kagome ice reported by Melko et al appears in two lobes at $h > 0$ and moderate $J_{\pm\pm}/J_z$. Translated into our parameters, this would correspond to a finite $J_x = +J_{\pm\pm}$ and $J_y = -J_{\pm\pm}$. The FM ferromagnetic phase, however, may be reminiscent of our experimental 3 in – 3 out magnetic order observed within the the kagome plane.

So, we agree with the referee when he/she writes: “The exchange parameters for this phase are not consistent with Nd₂Zr₂O₇”. But there is no contradiction here at all and this cannot be considered as a problem with our work. We do, however, want to cite the model of Melko et al. to put our study in a broader context.

We have changed the title and reworded the related sentences in the text to make this point clearer, and avoid any confusion for the reader between quantum kagome ice and dynamic kagome ice states.

The authors continue their response with:

“We should also point out that even if the breakdown of spin wave excitations is indeed well established from an experimental point of view in some frustrated pyrochlore systems, to our knowledge, no theoretical model has been published to reproduce such observations, so that the mechanisms at play are not elucidated yet. In that sense, the general character of such unusual inelastic spectra, that are also observed here in Nd₂Zr₂O₇, is an important point for the general understanding of the geometrically frustrated systems in the presence of quantum effects.”

Again, I will agree that there are unusual excitations here. However, extraordinary claims need extraordinary evidence. Without a model that fits your data it is difficult to claim that these excitations are solely from a dynamic kagome state.

In this paper, the main result is that the [111] field gives rise to a reduction of the dimensionality of the zero field “spin ice like” mode observed in Nd₂Zr₂O₇ at $H = 0$. Interestingly this mode becomes “kagome ice like” in finite [111] field. This evolution resembles the evolution of the diffuse scattering in canonical spin ice to kagome ice in a [111] field. However, and this is precisely the novelty of our work, our observations hold for the excitation spectrum. To the best of our knowledge, such a feature has not been reported to date, either experimentally or theoretically. In

addition, as already explained above, our theoretical calculations, carried out in the framework of the model proposed by Benton (and outlined in our own work, Nature Physics 12, 746-750 (2016)), are in line with this view and reproduce the main features of our data.

We however do not deny that there are discrepancies when comparing the data and the outcome of the model. The broadening of the spectra is indeed an issue, but it is not possible to solve it at the level of a spin wave approach. By the way, we note that the theoretical work by O. Benton does not solve this problem either. We emphasize that this point is not ignored in the text but is already discussed. However, to better explain our point, we have reworded the corresponding paragraph in our manuscript, giving more insight into the possible mechanisms that may explain this broadening. Indeed, as pointed by Benton in his paper, the model involves divergence free and divergence full excitations that, at the level of a spin wave approach, appear decoupled. We think that this exact decoupling is no more valid beyond the mean field approximation, which may lead to interactions between the modes, giving rise to significant broadening.

Finally, we do not argue that anything is extraordinary here, and we just compare our results with previous observations on other systems.

The authors continue with:

“To extend the discussion, we would like however to point to strong differences between Nd₂Zr₂O₇ and the two systems mentioned by the referee, Yb₂Ti₂O₇ and Er₂Pt₂O₇. First, it is important to note that the nature of the magnetic moment itself is very different, since the Nd ion has an Ising dipolar octupolar doublet, while Er and Yb ions have an XY anisotropy, which considerably modifies the spin dynamics. Second, in Er₂Pt₂O₇, only one study in zero field on a powder sample has been reported which provided no evidence of spin waves, maybe due to the proximity to competing phases. An anisotropy gap is present in the ordered state, as predicted theoretically for the Palmer-Chalker phase. It should be noted that a study of the related compound Er₂Sn₂O₇ shows well defined spin waves in the field polarized phase, even on a powder sample.

Third, in Yb₂Ti₂O₇, for which more experimental studies are available, and where the role of several competing phases has also been evoked, well defined spin waves are only observed in a high magnetic field, together with the formation of a two magnon continuum. The zero field spectrum does not present well defined excitations at all.

In Nd₂Zr₂O₇, the flat mode and the associated spin waves are rather well-defined in the whole field range: although the features are broader than usual spin wave excitations (a weak continuum of excitations likely exists to some extent), the neutron data clearly feature a dispersion, similar in the q-space as the model predictions, suggesting that the concept of spin wave quasi-particle is still valid. Some complex processes involving scattering of several quasi-particles are likely at play in the spin dynamics, but do not affect the main picture of a kagome ice mode.”

I agree that the physics is different in the other pyrochlore systems. I was just pointing out that quasiparticle breakdown is not a unique phenomena, nor a smoking gun for dynamic kagome ice states.

As we have already said in our first reply, we have never argued that quasiparticle breakdown is a unique feature of our system.

With respect to the comments about “flat modes” in Nd₂Zr₂O₇: the authors point out two different perspectives about these. In one perspective, the modes are due to the lifting of the spin doublet (page 6). Later on the paper (page 9), the authors claim that the excitations are rather robust to applied fields. How can both perspectives be present? If the modes are lifted doublets, does that not indicate that they are affected by applied fields (as they should be?).

Here again it seems likely that the referee is confusing two separate issues: on page 6, and written explicitly in the text, we are discussing the flat high-energy mode which reflects the dynamics of the apical spin. As it is frozen out by the [111] field, its dynamics are captured in a flat mode whose energy increases with the applied field. This mode forms from the zero field dispersive branch. In contrast, on page 9, we discuss the low-energy “two-dimensional kagome ice mode” which forms from the low-energy “spin ice mode”. We discuss its remarkable stability, since it persists up to about 1 T.

We have modified the manuscript to clarify this point and the difference between the two types of excitations we are discussing.

In the absence of a model for the data, the paper reports six-fold symmetry inelastic neutron scattering features in applied magnetic fields for Nd₂Zr₂O₇. The rest is unfortunately speculation. I agree that it is likely that theory cannot explain the data. This paper could be justified, if the editor sees fit, as a motivation for theorists to tackle the problem. At the moment, though, I find the results messy and difficult to interpret beyond the discovery of a new broadened excitation, which could be due to quasiparticle breakdown or some more exotic physics. There simply isn't enough data or theory to solidify the main thesis of the work.

We reiterate that our model explains the main features of the data. In addition, it is an unfair exaggeration to reduce our work to the observation of a broad excitation. While this broadness is a relevant experimental feature that is discussed in the text, our point is, again, that the field gives rise to a reduction of the dimensionality of the zero field “spin ice like” mode.

Furthermore, in this comment, the referee suggests that we do not have enough data. We would like to underline, first, that Referee 3 has a completely different opinion; as a matter of fact, we present magnetization, neutron diffraction, inelastic neutron scattering data and RPA calculations, providing a comprehensive description of the ground state and of the spin dynamics of Nd₂Zr₂O₇ in a [111] applied field.

(2) In my last point of my first review:

As the authors point out, there are other significant issues with the spin wave calculations. The field induced transition at $H = 0.08$ T cannot be accounted for. The high energy mode is shifted to higher fields than the model. And the lack of pinch points in the data is not accounted for in the model. The latter point is easier to explain. However, the first two points should be elucidated

in more detail before publication.

The authors respond with:

“Answer: as pointed out by the referee (and written explicitly in the manuscript), the mean field and spin wave approach to the XYZ Hamiltonian cannot account for the value of the metamagnetic field, the field dependence of the high energy mode, as well as the blurring of the pinch points, and more generally the width of the observed inelastic features. We have of course tried other analysis to account for these observations, especially the first two points.”

The authors then discuss their approaches (which do not work), and suggest that this paper might stimulate future theoretical work. I am not opposed to this line of reasoning as justification for the publication of this work. However, I do think that Nature publications are, in general, very rigorous and that experimental work submitted for publication should be accompanied by theoretical backing.

First, we would like to kindly draw attention to the fact that we do provide such a theoretical backing, and believe that it is unreasonable to demand that this theoretical discussion closes every open question and explains all the new physics revealed by our experimental data. Furthermore, we would like to highlight the fact that a significant number of papers in the field of frustrated magnetism including those published in Nature journals, have reported novel experimental phenomena without providing a full theoretical understanding (see e.g., Mirebeau *et al.* Nature **420**, 54 (2002), Kimura *et al.*. Nature Comm. **4**, 1934 (2013), Pan *et al.*. Nature Physics **12**, 361 (2016) Sibille *et al.* Nature Comm. **8**, 892(2017), Kermarrec *et al.*, Nature Comm. **8**, 14810 (2017),...).

In many pyrochlores, it is known that spin wave theory breaks down for certain parts of the phase diagrams, and this is what drives many physicists to study these compounds. However, in this case, the experimentalists have a sixfold smeared out inelastic scattering pattern at low temperatures in Nd₂Zr₂O₇ in an applied field. The sixfold symmetry of their pattern is anticipated (or at least not unlikely) from previous work. The smearing is not, but it is not entirely unexpected given that other pyrochlores have this behaviour (for various reasons). What is the new physics here, then?

We contend that the criticism about the novelty is unreasonable. While the reduction of the dimensionality of the correlations has already been discussed in the presence of an applied field in pyrochlores, to our knowledge, nobody has predicted nor even discussed the fact that the excitations, and in our case the dynamic spin ice mode, would follow the same picture as the correlations, i.e. a dimensional reduction and a dynamic kagome ice mode.

Without the proper J constants, this work cannot be mapped onto what O. Benton and others are modelling for the Nd pyrochlores. How can this be placed on their (rich) phase diagrams? Recent papers by Melko and others are also at odds with this work.

Actually, our analysis is precisely made within the framework of O. Benton’s model and our refined parameters are close to the ones determined by this author. Several sets of J parameters have been shown to fit the zero-field data but our measurements in applied field add constraints which allow us to better refine these parameters (as detailed in the supplementary information).

We thus do not understand a comment like: “Without the proper J constants, this work cannot be mapped onto what O. Benton and others are modeling for the Nd pyrochlores.”

As already mentioned, the recent papers by Melko et al. address a different issue, and we reiterate that there is no contradiction with our work. They discuss a peculiar ground state, the *quantum kagome ice* state, that appears in a different part of the phase diagram resulting from the XYZ Hamiltonian that we (and Benton) use.

Minor points:

(1) In the last sentence of the abstract, there is a spelling error (manifests) and the sentence is not grammatically correct.

(2) On page 9, there is an awkward statement about “the broadness of the feature reminds observation...” This should be reworded.

We have corrected these two points.

Reviewer 3

The research is of excellent experimental quality and the ideas presented are highly original. Certainly this paper will influence the direction of the field.

The revised version is much clearer and throws into sharp relief the key observations and claims. Application of field along 111 disconnects the apical spin producing what could be classical kagome ice patterns in plane; there is good experimental evidence for this. A new flat dynamical mode also appears which is attributed to quantum dynamics of the out of plane component of the same spins whose in plane components appear to be near classical kagome ice. This is an intriguing hypothesis but is not proven definitively.

I support publication of this fascinating paper which is now incisive, compelling and important - even if some points lack iron-clad proof.

We thank the referee for these encouraging comments. The evidence for the dimensionality reduction is quite clear from the data, along with the rise of the new set of dispersive branches, but we acknowledge that our present understanding should evolve in the future, calling for further theoretical studies.

Reviewers' Comments:

Reviewer #1 (Remarks to the Author):

Dear editor:

I have had the opportunity to evaluate the revised manuscript "Dynamic kagome ice" by Lhotel et al. The manuscript has improved to the point where it can be considered to be published in Nature Communications. I remain skeptical about the "novelty" of the results, and the interpretation of the "dynamic ice phase", but the results are still important and should be communicated within the general community.

I had the following last issues to be addressed before publication:

(1) The authors claim on page 3 that "Mean field calculations.... reproduce qualitatively our observations." This sentence should be deleted. The only calculation that represents the inelastic scattering data in Figure 4 is panel (a) in zero applied field. The rest of the calculations do not agree, which I think is one of the points of the work. The authors admit this in point (1) of their reply letter where they state that "calculations of zero field match the data." The rest of the calculations are not a good match.

(2) Error bars should be listed for the revised J parameters on page 8, at the end of the second paragraph. This is the only way that model compounds and theory can be compared.

(3) One last comment on the author's claim of "novelty." Novelty is one of the requirements for publication in Nature journals. However, a six fold, low energy inelastic response in a pyrochlore system with applied fields perpendicular to kagome planes is not novel. The interpretation might be novel in this case, but the actual observation of such a pattern has been made before. Yb₂Ti₂O₇ in applied fields along the <111> has a similar response, for example (K. A. Ross et al, PRB (2011)). However, there is not a dynamic kagome ice state in Yb₂Ti₂O₇. The authors have wisely not used this language of novelty much in the actual paper. However, claims such as "this is the first time that this physics is observed experimentally" (page 1 of their reply) and "nobody has predicted .. the correlations" (page 5 of their reply) should be made with caution. Weak, inelastic scattering has been observed in kagome and pyrochlore systems before, and some of the physics is very similar to what is claimed here. While I am skeptical of these results, I can see the value of the larger physics community to have the opportunity to evaluate this work in more detail.

Finally, I would like to comment on one other response that the authors have made - that referee

3 completely agrees with their conclusions. This is not true - Referee 3 clearly comments that "some points lack iron-clad proof" and "This is an intriguing hypothesis but is not proven definitively." Referee 3 also has doubts about their work. It is not honest to claim as the authors do on page 4 of their reply, that "Referee 3 has a completely different opinion." However, my job as a reviewer is not to comment on the ethics of the authors, but only to look at the arguments made within the manuscript. I am cautiously recommending publication in Nature Communications.

Reply to Reviewer

We have carefully read Reviewer 1's comments and have made the changes addressing this reviewer's remarks. We present below a point-by-point response to the views expressed by the referee.

Reviewer #1 (Remarks to the Author):

Dear editor:

I have had the opportunity to evaluate the revised manuscript "Dynamic kagome ice" by Lhotel et al. The manuscript has improved to the point where it can be considered to be published in Nature Communications. I remain skeptical about the "novelty" of the results, and the interpretation of the "dynamic ice phase", but the results are still important and should be communicated within the general community.

I had the following last issues to be addressed before publication:

(1) The authors claim on page 3 that "Mean field calculations.... reproduce qualitatively our observations." This sentence should be deleted. The only calculation that represents the inelastic scattering data in Figure 4 is panel (a) in zero applied field. The rest of the calculations do not agree, which I think is one of the points of the work. The authors admit this in point (1) of their reply letter where they state that "calculations of zero field match the data." The rest of the calculations are not a good match.

We have removed this sentence. Nevertheless, we still believe that our calculations reproduce the main features of the inelastic spectra, even if not in a quantitative way.

(2) Error bars should be listed for the revised J parameters on page 8, at the end of the second paragraph. This is the only way that model compounds and theory can be compared.

The error bars were given in the supplemental information, together with the detailed fitting procedure. We have added the error bars in the main manuscript.

(3) One last comment on the author's claim of "novelty". Novelty is one of the requirements for publication in Nature journals. However, a six fold, low energy inelastic response in a pyrochlore system with applied fields perpendicular to kagome planes is not novel. The interpretation might be novel in this case, but the actual observation of such a pattern has been made before. Yb₂Ti₂O₇ in applied fields along the $\langle 111 \rangle$ has a similar response, for example (K. A. Ross et al, PRB (2011)). However, there is not a dynamic kagome ice state in Yb₂Ti₂O₇. The authors have wisely not used this language of novelty much in the actual paper. However, claims such as "this is the first time that this physics is observed experimentally" (page 1 of their reply) and "nobody has predicted .. the correlations" (page 5 of their reply) should be made with caution. Weak, inelastic

scattering has been observed in kagome and pyrochlore systems before, and some of the physics is very similar to what is claimed here. While I am skeptical of these results, I can see the value of the larger physics community to have the opportunity to evaluate this work in more detail.

We still do not agree with this point of view.

Even if some inelastic scattering has been observed in kagome and pyrochlore systems before, we are not aware of any example of such a six fold energy inelastic response in the presence of a magnetic field. In the example given by the referee (K. A. Ross et al. PRB 2011), the observed signal corresponds to a thermal evolution, and not to a field induced phenomenon. Moreover, it is an elastic response and not an inelastic one. In addition, even if rods are present, there is no six fold symmetry similar to the pattern we have observed. It was indeed argued that these correlations could be associated with the 2D kagome planes, but not to a kagome ice feature. This is consistent with the fact that, as quoted by the referee, there is no dynamic kagome ice state in $\text{Yb}_2\text{Ti}_2\text{O}_7$.

Finally, I would like to comment on one other response that the authors have made - that referee 3 completely agrees with their conclusions. This is not true - Referee 3 clearly comments that “some points lack iron-clad proof” and “This is an intriguing hypothesis but is not proven definitively.” Referee 3 also has doubts about their work. It is not honest to claim as the authors do on page 4 of their reply, that “Referee 3 has a completely different opinion.” However, my job as a reviewer is not to comment on the ethics of the authors, but only to look at the arguments made within the manuscript. I am cautiously recommending publication in Nature Communications.

In our response earlier, we were merely referring to the comment “The research is of excellent experimental quality and the ideas presented are highly original” made by Referee 3 which suggested a favourable judgement of our work. Finally, we would like to retain the meaningful and constructive comments and criticisms made by all the reviewers, which have helped us to improve the clarity of the manuscript.